# Characterisation of the Gut Bacteria of Cultured and Wild Spiny Lobster *Panulirus ornatus*

Mei C. Ooi *, Andrew J. Trotter, Gregory G. Smith and Andrew R. Bridle

Institute for Marine and Antarctic Studies, University of Tasmania, Launceston, TAS 7250, Australia
* Correspondence: mei.ooi@utas.edu.au

**Abstract:** The commercial onshore aquaculture of the spiny lobster *Panulirus ornatus*, while in its infancy, has progressed rapidly from the enabling research that continues at the University of Tasmania. The development of lobster feeds, both fresh and manufactured, has been critical to the success of this emerging aquaculture sector. Fresh feeds derived from mussel represent the gold standard in terms of the growth performance of juvenile lobsters. Nonetheless, concerns regarding availability, sustainability, and potential biosecurity issues of fresh feeds highlight the importance of developing manufactured feeds for lobster aquaculture. Wild lobsters are assumed to have a balanced natural diet that allows for standard growth and development, and as such natural diets are often used as a reference for feed development. Similarly, the gut microbiota associated with a natural diet is assumed to reflect a healthy microbial assemblage. The aim of this study was to compare the microbiota of the hindgut and hepatopancreas of cultured *P. ornatus* fed with a commercial prawn pellet or mussel to that of wild spiny lobster juveniles. Gut samples were analysed using Oxford Nanopore 16S rRNA gene sequencing. Based on principal coordinate analysis, the gut bacteria of cultured lobsters were different from the wild juveniles. The core microbiota of the hindgut and hepatopancreas libraries were phyla Proteobacteria (Gamma, Alpha) and Bacteroidetes. *Vibrio* was the most dominant genus in both organs. The differences in bacterial relative abundance were mainly between cultured (pellet-, mussel-fed) and wild lobsters. In conclusion, bacteria in the cultured lobsters had significantly different profiles to that of the wild juveniles, indicating that current onshore aquaculture practices alter the gut microbiota. A number of different feeding and culture practices may be required if the aim of closed culture practices is to attain a gut microbiota in cultured animals that is representative of that found in wild spiny lobsters.

**Keywords:** gut bacteria; wild spiny lobster; cultured spiny lobster; mussel; pellet; *Vibrio*



## 1. Introduction

Lobsters are a valuable commercial commodity, and the majority of world production originates from wild-caught fisheries with a minor component sourced from seedstock caught from the wild and grown-out in aquaculture. This latter practice places wild lobster stocks under considerable sustainability pressures. The commercial and ecological value of the ornate spiny lobster *Panulirus ornatus* continues to drive research to improve the closed-life cycle aquaculture of this species and hence reduce the fishing pressure on wild spiny lobster populations. The challenges of breeding lobsters are not trivial due to the complex and protracted larval phase, potential diseases, in addition to cannibalism [1,2]. Nevertheless, the production of the ornate spiny lobster *P. ornatus* from egg to adult was closed at the University of Tasmania. Commercial production has now commenced in Queensland, Australia.

The provision of appropriate nutritional requirements at different stages of lobster development are an important consideration in the growth and health of these animals. A commercial goal for the culture of *P. ornatus* in captivity is the development of a readily produced manufactured feed with a complete and consistent nutrient composition that

meets the needs of the lobster. One of the ways to understand the impact of feed is to examine the bacterial communities of the digestive system [3,4]. As the wild counterparts of a species are likely to have a balanced natural diet, these animals are often used as references including their gut microbial composition [5]. Juvenile wild spiny lobsters of the *Panulirus* genus have been reported to consume crustaceans, mollusks, fish, other animals and plants [6,7]. As these fresh feeds have microorganisms on or in them, in an aquaculture context it is generally considered safer to provide the nutrition via a manufactured feed, thereby minimising biosecurity concerns and the introduction of potential pathogens to the culture systems.

Herein we utilise long-read Oxford Nanopore Technology (ONT) DNA sequencing to examine microbial communities of the juvenile spiny lobster gut. DNA sequencing allows for complex bacterial compositions of the lobster gut to be characterised and taxonomically classified in a culture-independent manner. The advancement of ONT third generation sequencing allows for more accurate bacterial taxonomic identification using full length 16S rRNA gene reads compared to the inherent short reads produced by second generation sequencing. The digestive tract of lobster is composed of three sections, the foregut, midgut (including hepatopancreas/digestive gland) and hindgut, with each performing separate functions [8] and harbouring microorganisms [9]. In studies that examined the hindgut and hepatopancreas microbiota of wild and aquaculture reared Pacific white shrimp *Litopenaeus vannamei*, there were significant differences noted [3,4,10]. These differences in the gut microbiota are potentially important, as they are assumed to impact the growth performance and health of this and other cultured crustaceans [11].

With commercialisation and aquaculture production of *P. ornatus* comes a need for the development of a manufactured feed. A manufactured feed will not only reduce the dependence on fresh feed, where availability and quality vary over time, but will also minimise the potential for the introduction of pathogens and disease. Knowledge of the impacts of formulated feed on animal performance, including the impacts on gut microbiota, are necessary and will inform the large-scale commercial production of quality lobster feed. In this study, we taxonomically identify the hindgut and hepatopancreas microbiota of wild and cultured juvenile spiny lobsters, with the cultured lobsters fed on either a commercial prawn pellet or on fresh mussel.

## 2. Materials and Methods

### 2.1. Experimental Animals

Juvenile spiny lobsters were produced from hatch at the Institute for Marine and Antarctic Studies (IMAS), University of Tasmania, Australia. Newly emerged juveniles (J1) were held in 18 L tanks with a floor surface area of 0.09 m$^2$ at a stocking density of 110 m$^{-2}$ (i.e., 10 per tank), supplied with flow-through seawater. The seawater was filtered ($\leq$50 $\mu$m) and ozonated (350 mV). Water quality parameters during culture were: temperature 27.8 °C $\pm$ 0.3, pH 8.1 $\pm$ 0.1., salinity 34.1 $\pm$ 0.4 ppt, and dissolved oxygen 109.5 $\pm$ 7.0% sat. Newly emerged juveniles were fed twice daily, with moist feed (IMAS commercial-in-confidence manufactured moist feed) until 5 days post-emergence. The lobsters were then weaned onto the treatment feeds (blue mussel *Mytilus galloprovincialis* or kuruma prawn pellet [Higashimaru, Japan]) over 4 days by feeding a mixture of treatment feeds and the IMAS feed. Subsequently, only treatment feeds were provided for 72 days of culture. Lobsters were fed twice daily (morning and afternoon) to excess adjusted based on visual observation. Vessels were siphoned clean, and uneaten feeds were removed before new feeds were provided, and all cleaning and siphon apparatus were chlorinated between tanks. Seven mussel-fed lobsters (3.9 $\pm$ 0.7 g) and six pellet-fed lobsters (1.3 $\pm$ 0.5 g) of the same age were sampled at the end of the treatment period.

Wild spiny lobsters were collected from the Cairns Inlet, North Queensland, Australia, by divers. The lobsters were removed from the jetty pylons and transported live to the IMAS facility within 24 h of capture. These animals were used for gut microbiota comparison with cultured animals. Six wild juveniles (0.8 $\pm$ 0.1 g) were sampled upon arrival in Tasmania.

The species was confirmed as *Panulirus ornatus* via Sanger sequencing of the cytochrome c oxidase I gene. The PCR program was 95 °C for 3 min; 40 cycles of 95 °C for 10 s, 53 °C for 30 s, and 72 °C for 15 s. All of the sampled cultured and wild juveniles were in intermoult phase except for three wild animals which were in premoult stage.

### 2.2. Sample Collection

The juvenile lobsters were euthanised in a seawater ice slurry. Aseptic techniques were practiced during lobster dissection including sterilising surgical tools using 1:20 chlorine 4% *w/v* and 70% ethanol. An inverted Y incision was made on the dorsum of the cephalothorax to sample the hepatopancreas. After incising the tail fan, the hindgut was pulled out via the detachment of the cephalothorax. The samples were placed in nucleic acid preservation solution (4 M ammonium sulphate, 25 mM sodium citrate, 10 mM EDTA; pH 5.5) at 4 °C overnight prior to longer-term storage at −20 °C.

### 2.3. Total Nucleic Acids (TNA) Extraction

The hindgut was briefly rinsed in LC-MS grade water (LiChrosolv, Merck, Germany) to minimise the carryover of preservation solution that may affect the extraction of TNA. The hindgut was lysed in 300 μL of urea extraction buffer (4 M urea, 0.5% sodium dodecyl sulphate, 0.2 M sodium chloride, 10% glycerol) and 0.13 μg μL$^{-1}$ of Proteinase K (Bioline, Narellan, NSW, Australia) at 55 °C for 35 min. After cooling the lysate on ice for 10 min, 175 μL of 7.5 M ammonium acetate was added, vortexed for 30 s and centrifuged at 14,000× *g* for 5 min at 18 °C. The supernatant was mixed with 2 μL of polyacryl carrier (Molecular Research Center, Cincinnati, OH, USA) and 500 μL of isopropanol with 0.02 μg μL$^{-1}$ pink co-precipitant (Bioline), incubated at room temperature for 10 min and centrifuged at 16,000× *g* for 10 min. The pellet was rinsed with 500 μL of 60% ethanol twice, air dried, and resuspended in 50 μL of LC-MS grade water.

The same TNA extraction steps were used for the hepatopancreas except that the isopropanol with pink co-precipitant was replaced with polyethylene glycol (20% PEG 8000, 1.2 M NaCl, 10 mM MgCl$_2$, 4 mM TRIS) and centrifuged at 12,000× *g* for 15 min. Five samples were extracted for each treatment group for each organ. Two negative controls (extraction reagents only) were included.

### 2.4. PCR and Sequencing

For the hindgut, the 50 μL PCR reactions contained 25 μL of LongAmp Taq 2X Master Mix (New England Biolabs, Ipswich, MA, USA), 1 μL of 16S rRNA gene barcoded primer (16S Barcoding Kit SQK-RAB204, Oxford Nanopore Technologies, Littlemore, Oxford, UK), and 6 μL of 1:10 diluted TNA extract. The PCR program used on a C1000 Thermal Cycler (Bio-Rad Laboratories Inc., Irvine, CA, USA) was 95 °C for 1 min; 36 cycles of 95 °C for 20 s, 55 °C for 30 s, 65 °C for 2 min; and 65 °C for 5 min.

For the hepatopancreas, a nested PCR was used to increase DNA quantity. The primary PCR mixture (20 μL reaction) comprised 10 μL of 2× MyTaq HS mix, 500 nM each of 27F (5′-AGAGTTTGATCMTGGCTCAG-3′) and 1492R (5′-TACGGYTACCTTGTTACGACTT-3′) 16S rRNA gene primers, and 1 μL of 1:10 diluted TNA extract. The primary PCR program was 95 °C for 3 min, 20 cycles of 95 °C for 15 s, 50 °C for 30 s, 72 °C for 45 s; and 72 °C for 3 min. One microliter of the primary PCR product was used in secondary PCR, which consisted of 25 μL of 2× MyTaq HS mix, 10 μL of 16S rRNA gene barcoded primer (16S Barcoding Kit 1-24 SQK-16S024), and 14 μL of microbial DNA-free water (Qiagen, Germantown, MD, USA). The secondary PCR program was 95 °C for 1 min, 25 cycles of 95 °C for 15 s, 55 °C for 30 s, 72 °C for 60 s, and 72 °C for 3 min.

The following steps were taken according to the 16S Barcoding Kit protocol. Secondary PCR products were purified using AMPure XP beads (Beckman Coulter, Indianapolis, IN, USA) and quantified using a Qubit fluorometer (Invitrogen, Life Technologies Australia, Melbourne, VIC, Australia). Eluates were pooled to make libraries and were loaded onto the SpotON R9.4.1 flow cell in a MinION sequencer.

*2.5. Data Analyses*

The long-read sequences were basecalled using the ONT software Guppy v5.0.16 in super accuracy mode, filtered with a minimum q score of 8, demultiplexed and trimmed. The bacterial taxonomy was assigned to the sequences using Emu v2.0.1, which combined the rrnDB and NCBI databases [12]. Bacteria identified in the negative controls were removed from the samples. A table containing assigned reads of each sample was uploaded to MicrobiomeAnalyst [13] to examine alpha diversity (Observed features, Shannon index, Simpson's Index of Diversity [1-D]), beta diversity, relative abundance and core microbiota (sample prevalence $\geq$ 20%, relative abundance $\geq$ 0.01%). Data were filtered to a minimum count of 2 with 10% prevalence in the sample and normalised by rarefaction to the minimum library size (i.e., 14,485 for the hindgut and 8182 for the hepatopancreas) and total sum scaling. The beta diversity was analysed by Bray Curtis distance-based principal coordinate analysis at species level and PERMANOVA. Feature relative abundance among treatment groups were compared using RNAseq (DESeq2), and an adjusted *p* value of <0.05 was considered significant. The raw sequences were deposited in NCBI SRA under BioProject accession number PRJNA821626. To show shared and unique bacterial genera among treatment groups, Venn diagrams were produced using InteractiVenn [14].

## 3. Results

*3.1. Overview of the Hindgut and Hepatopancreas Microbiota*

There were no significant differences in the alpha diversity indices ($p > 0.05$) including observed features, Shannon, and Simpson among three treatment groups in the hindgut and hepatopancreas libraries, respectively (Table 1). The first two axes of principal coordinate analysis based on the Bray Curtis distance matrix explained 67.1% and 52.1% of the variation in the abundance of features among the hindgut and hepatopancreas libraries, respectively (Figure 1). The treatment groups within the hindgut ($R^2 = 0.350$, $p = 0.009$) and hepatopancreas libraries ($R^2 = 0.352$, $p = 0.002$) were significantly different when analysed statistically using PERMANOVA. The cluster of hepatopancreas libraries of wild lobsters was separated from the pellet-fed juveniles, showing dissimilarity between wild and cultured animals (Figure 1B).

*3.2. Hindgut Microbial Community Composition*

The core microbiota of the hindgut libraries consisted of Proteobacteria and Bacteroidetes, which were the most represented phyla. The hindgut libraries of wild lobsters showed the highest Proteobacteria to Bacteroidetes ratio at 122.3, followed by mussel-fed animals at 38.5 and pellet-fed juveniles at 18.2. There was significantly less representation ($p < 0.05$) of Firmicutes but more Fusobacteria in the hindgut of pellet-fed lobsters (Firmicutes: 0.1 $\pm$ 0.1%, Fusobacteria: 1.2 $\pm$ 0.4%) compared to the mussel-fed (Firmicutes: 1.5 $\pm$ 0.8%, Fusobacteria: 0%) and wild animals (Firmicutes: 3.6 $\pm$ 3.0%, Fusobacteria: 0%).

Within phylum Proteobacteria, the classes Gammaproteobacteria and Alphaproteobacteria dominated the hindgut libraries. The relative abundance of the classes Bacilli and Fusobacteriia belonging to phyla Firmicutes and Fusobacteria, respectively, were significantly different ($p < 0.05$) among gut libraries with the same trend at the phylum level. The class Deltaproteobacteria was present ($p < 0.01$) in the hindgut of pellet- (1.4 $\pm$ 1.0%) and mussel-fed juveniles (0.2 $\pm$ 0.1%), but absent in wild lobsters.

The top five bacterial families in the hindgut based on relative abundance were *Vibrionaceae*, *Roseobacteraceae*, *Cellvibrionaceae*, *Oceanospirillaceae* and *Rhodobacteraceae*. The families *Cohaesibacteraceae* and *Spongiibacteriaceae* were present ($p < 0.01$) in the hindgut of pellet (*Cohaesibacteraceae*: 1.8 $\pm$ 1.3%, *Spongiibacteriaceae*: 1.4 $\pm$ 0.6%) and mussel-fed lobsters (*Cohaesibacteraceae*: 0.3 $\pm$ 0.2%, *Spongiibacteriaceae*: 1.5 $\pm$ 1.3%), but absent in the wild animals. Conversely, the family *Carnobacteriaceae* was not found in the hindgut of pellet-fed juveniles but present ($p < 0.01$) in the wild (1.2 $\pm$ 1.0%) and in mussel-fed lobsters (1.2 $\pm$ 0.6%).

**Table 1.** Alpha diversity indices for normalised gut libraries of juvenile *P. ornatus*.

| Lobster ID | Raw Read Counts | Observed Features | Shannon | Simpson |
|---|---|---|---|---|
| | | Hindgut of wild lobsters | | |
| W3 | 130,020 | 74 | 1.28 | 0.41 |
| W4 | 65,478 | 77 | 1.86 | 0.69 |
| W5 | 14,485 | 17 | 0.99 | 0.44 |
| W6 | 32,123 | 25 | 0.68 | 0.24 |
| W7 | 57,126 | 55 | 2.10 | 0.74 |
| | | Hindgut of pellet-fed lobsters | | |
| K1A1 | 68,558 | 112 | 2.33 | 0.72 |
| K1A2 | 104,684 | 121 | 2.68 | 0.86 |
| K2A1 | 62,317 | 73 | 0.93 | 0.28 |
| K2A2 | 38,106 | 80 | 2.10 | 0.70 |
| K3A1 | 67,613 | 109 | 2.97 | 0.87 |
| | | Hindgut of mussel-fed lobsters | | |
| M1A1 | 22,441 | 79 | 2.78 | 0.85 |
| M1A3 | 42,545 | 36 | 0.57 | 0.17 |
| M2A2 | 81,850 | 160 | 3.09 | 0.86 |
| M2A3 | 30,128 | 53 | 2.14 | 0.76 |
| M2A4 | 84,808 | 68 | 2.26 | 0.77 |
| | | Hepatopancreas of wild lobsters | | |
| W2 | 29,673 | 96 | 3.51 | 0.93 |
| W3 | 54,531 | 82 | 2.93 | 0.86 |
| W4 | 38,066 | 69 | 2.98 | 0.92 |
| W5 | 65,152 | 58 | 2.82 | 0.90 |
| W7 | 80,957 | 13 | 0.66 | 0.35 |
| | | Hepatopancreas of pellet-fed lobsters | | |
| K1A1 | 69,171 | 138 | 3.01 | 0.80 |
| K1A2 | 23,137 | 53 | 1.89 | 0.65 |
| K2A1 | 23,922 | 58 | 3.16 | 0.94 |
| K2A2 | 8388 | 32 | 1.85 | 0.62 |
| K3A5 | 75,792 | 118 | 3.36 | 0.92 |
| | | Hepatopancreas of mussel-fed lobsters | | |
| M1A1 | 43,208 | 26 | 0.84 | 0.31 |
| M1A3 | 84,819 | 17 | 0.76 | 0.31 |
| M1A4 | 8182 | 14 | 1.10 | 0.43 |
| M2A4 | 18,198 | 30 | 1.02 | 0.34 |
| M3A3 | 20,321 | 46 | 2.71 | 0.87 |

The five most dominant bacterial genera in the lobster hindgut libraries were *Vibrio*, *Ruegeria*, *Pesudoteredinibacter*, *Shimia* and *Aliiroseovarius* (Figure 2). Six genera were relatively more abundant ($p < 0.05$) in the hindgut of pellet- (*Cohaesibacter*: $1.8 \pm 1.3\%$, *Neptuniibacter*: $1.1 \pm 0.3\%$, *Roseovarius*: $1.9 \pm 1.7\%$, *Ruegeria*: $4.3 \pm 2.2\%$, *Shimia*: $1.1 \pm 0.5\%$, *Spongiibacter*: $1.3 \pm 0.6\%$) and mussel-fed animals (*Cohaesibacter*: $0.3 \pm 0.2\%$, *Neptuniibacter*: $1.3 \pm 1.0\%$, *Roseovarius*: $0.1 \pm 0.1\%$, *Ruegeria*: $2.0 \pm 1.8\%$, *Shimia*: $4.1 \pm 1.9\%$, *Spongiibacter*: $1.5 \pm 1.2\%$) compared to their wild counterparts. Among the three treatment groups, three genera: *Atopobacter*, *Oleispira* and *Phaeobacter*, were relatively less abundant ($p < 0.05$) in the hindgut of pellet-fed lobsters. All nine genera mentioned above also showed significant differences among treatment groups at the species level. Interestingly, several *Vibrio* species were relatively more abundant ($p < 0.05$) in the hindgut of mussel- (*Vibrio owensii*: $20.3 \pm 8.1\%$, *Vibrio hyugaensis*: $1.7 \pm 1.2\%$, *Vibrio jasicida*: $0.7 \pm 0.4\%$) and pellet-fed lobsters (*Vibrio owensii*: $4.4 \pm 2.2\%$, *Vibrio hyugaensis*: $16.6 \pm 6.6\%$, *Vibrio jasicida*: $12.4 \pm 9.6\%$) compared to the wild animals.

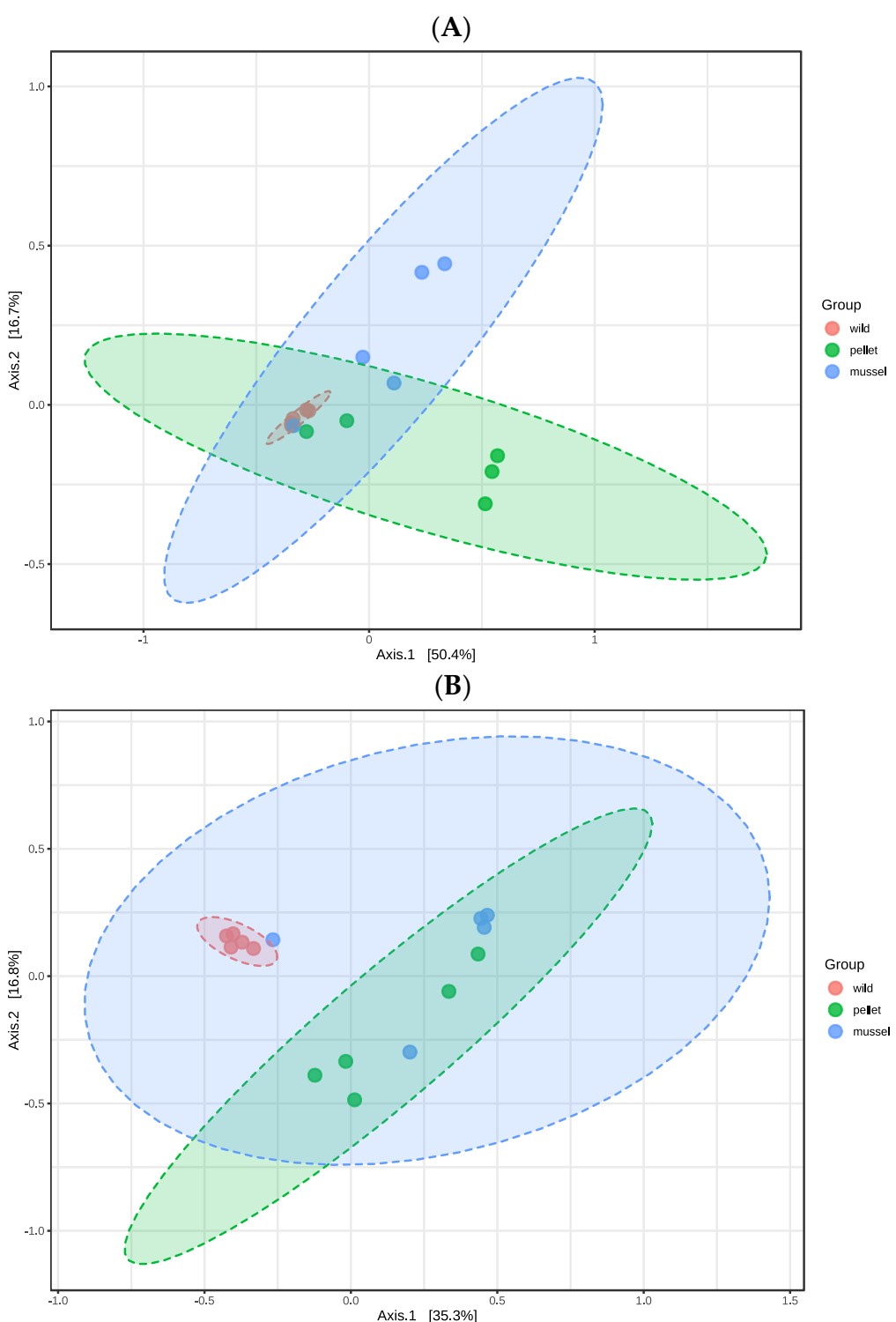

**Figure 1.** Principal coordinate analysis plots showing similarity in the (**A**) hindgut (*p* = 0.009) and (**B**) hepatopancreas sequence libraries (*p* = 0.002) of *P. ornatus* at species level.

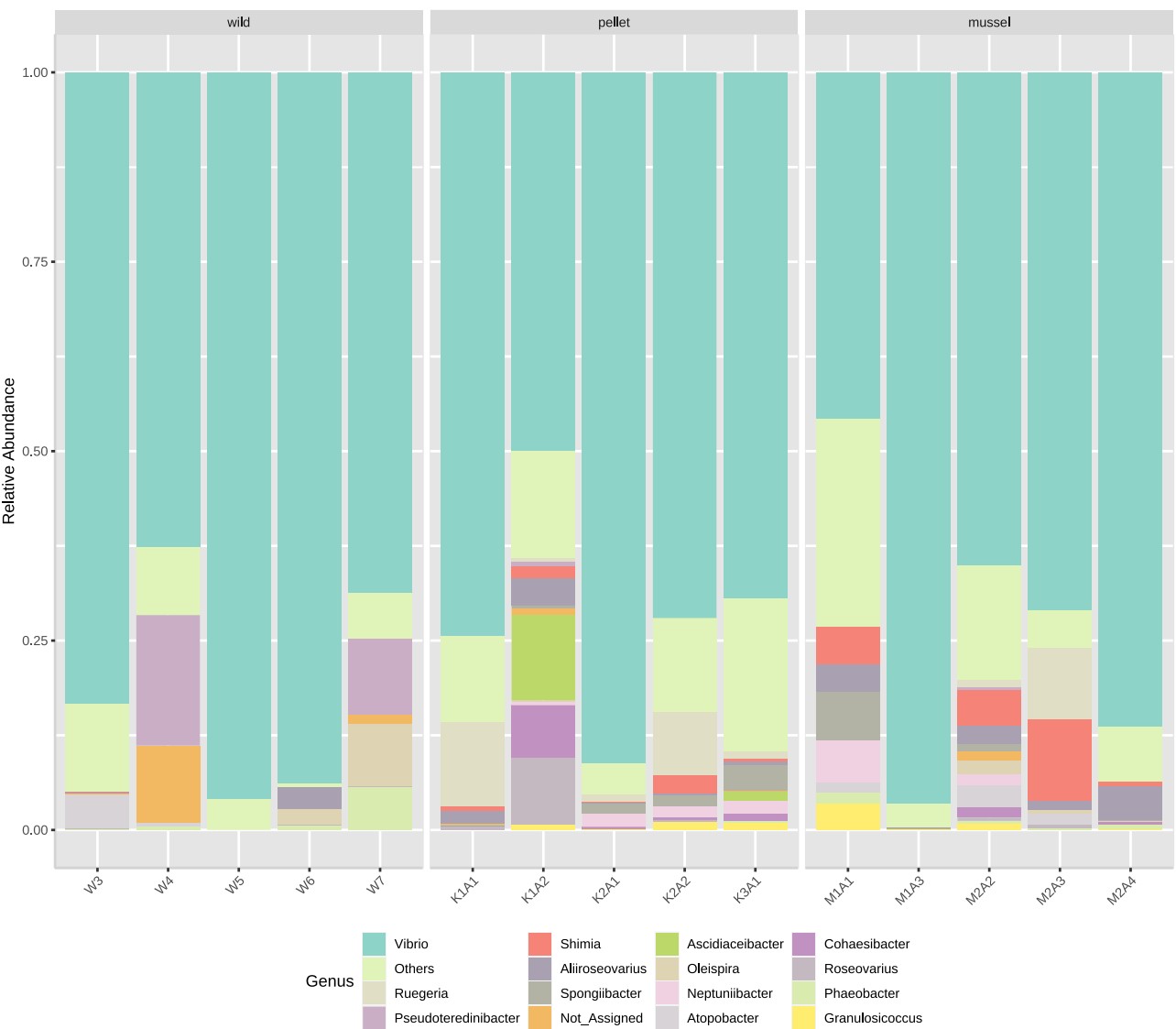

**Figure 2.** Relative abundance of bacterial features in the hindgut libraries of juvenile *P. ornatus* at genus level.

### 3.3. Hepatopancreas Microbial Community Composition

The core microbiota of the hepatopancreas libraries were phyla Proteobacteria and Bacteroidetes. The highest to lowest Proteobacteria to Bacteroidetes ratios were from the hepatopancreas libraries of wild juveniles at 21:7, mussel-fed lobsters at 14:0 and pellet-fed animals at 8:3. Phylum Tenericutes was relatively more abundant ($p < 0.01$) in the hepatopancreas of wild lobsters ($1.7 \pm 1.4\%$) than in the cultured animals. Nevertheless, the hepatopancreas of pellet-fed juveniles had significantly more ($p < 0.05$) phylum Planctomycetes ($1.3 \pm 0.8\%$) compared to the other two groups.

The three most abundant classes in the hepatopancreas libraries were Gammaproteobacteria, Alphaproteobacteria and Bacilli. The classes Mollicutes (phylum Tenericutes) and Planctomycetia (phylum Planctomycetes) showed similar trends of significant differences ($p < 0.05$) at the phylum level. The class Deltaproteobacteria was relatively more abundant ($p < 0.05$) in the hepatopancreas of pellet- ($1.4 \pm 0.7\%$) and mussel-fed lobsters ($0.8 \pm 0.6\%$) compared to the wild juveniles. Conversely, class Erysipelotrichia was only detected ($p < 0.01$) in in the hepatopancreas of wild animals ($2.1 \pm 1.5\%$).

The five most common families in the hepatopancreas libraries were *Vibrionaceae*, *Roseobacteraceae*, *Alteromonadaceae*, *Thiotrichaceae* and *Rhodobacteraceae*. The hepatopancreas

of wild lobsters had more ($p < 0.01$) families *Alteromonadaceae* (14.8 ± 7.2%) and *Carnobacteriaceae* (3.9 ± 3.1%) than the cultured juveniles. In contrast, the families *Thiotrichaceae* and *Methylobacteriaceae* were relatively more abundant ($p < 0.01$) in the hepatopancreas of pellet-fed animals (*Thiotrichaceae*: 8.6 ± 3.9%, *Methylobacteriaceae*: 2.8 ± 2.2%), but less abundant in the mussel-fed lobsters (*Thiotrichaceae*: 5.0 ± 5.0%, *Methylobacteriaceae*: 0.7 ± 0.7%) and absent in the wild juveniles.

The top four most abundant genera in the hepatopancreas libraries were *Vibrio*, *Alteromonas*, *Thiothrix* and *Ruegeria* (Figure 3). Genera *Alteromonas* and *Atopobacter* were found ($p < 0.01$) in the hepatopancreas of wild lobsters (*Alteromonas*: 14.8 ± 7.2%, *Atopobacter*: 3.5 ± 2.8%) but not in the cultured animals. Two genera, *Methylobacterium* (family *Methylobacteriaceae*) and *Thiothrix* (family *Thiotrichaceae*), showed the same trend at the family level. *Amylibacter* and *Marinicella* were two additional genera that were not detected in the hepatopancreas of wild animals but present ($p < 0.01$) in the mussel- (*Amylibacter*: 2.9 ± 0.9%, *Marinicella*: 1.3 ± 0.8%) and pellet-fed lobsters (*Amylibacter*: 0.8 ± 0.3%, *Marinicella*: 1.7 ± 1.1%). All six genera stated above also had species with significant differences among treatment groups. Furthermore, several species of *Vibrio* were relatively more abundant ($p < 0.05$) in certain treatment groups, such as *Vibrio owensii*, which dominated the hepatopancreas of mussel-fed lobsters (mussel-fed: 52.1 ± 16.2%, pellet-fed: 19.6 ± 12.1%, wild 0.1 ± 0.1%). *Vibrio jasicida* and *Vibrio hyugaensis* were relatively more abundant in the hepatopancreas of pellet-fed juveniles (*V. jasicida*: 12.8 ± 12.2%, *V. hyugaensis*: 5.6 ± 3.1%) compared to the mussel-fed and wild lobsters. There was more *Vibrio campbellii* in the hepatopancreas of wild animals (16.8 ± 15.5%) than the pellet- (0.6 ± 0.4%) and mussel-fed juveniles (0%).

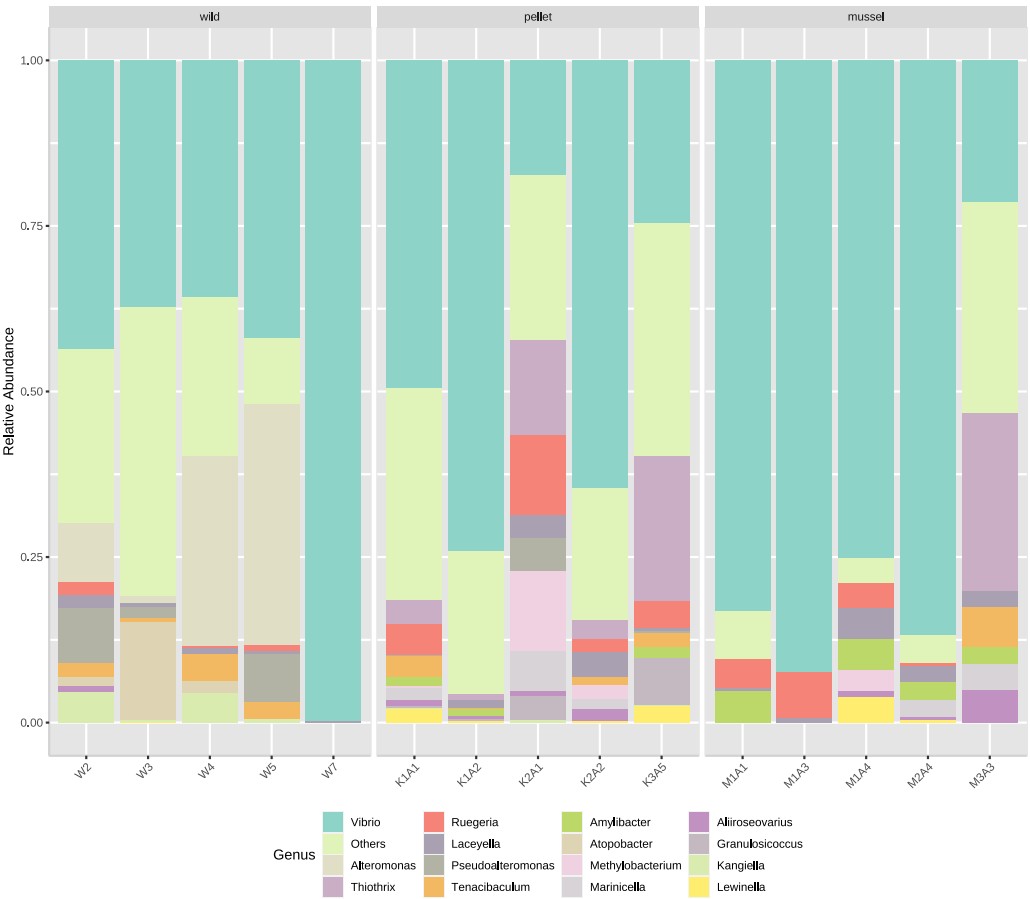

**Figure 3.** Relative abundance of bacterial features in the hepatopancreas libraries of juvenile *P. ornatus* at genus level.

### 3.4. Shared and Unique Bacterial Genera in the Hindgut and Hepatopancreas

Genera *Vibrio* and *Ruegeria* were part of the core microbiota of both the hindgut and hepatopancreas (Figure 4). The Venn diagrams showed that 30 bacterial genera (44% of wild, 24% of mussel-fed, and 24% of pellet-fed) were shared by the three hindgut libraries (Figure 4A). The three hepatopancreas libraries shared 14 genera of bacteria (27% of mussel-fed, 22% of wild, and 12% of pellet-fed) (Figure 4B). Out of both organs, the genera *Vibrio*, *Thalassobius*, *Thalassococcus* and *Aliiroseovarius* were found. Both the hindgut and hepatopancreas libraries of pellet-fed animals had the highest number of bacterial genera and shared more bacteria with mussel-fed juveniles compared to the wild. The lowest number of unique bacterial genera was found in the libraries of both organs of mussel-fed lobsters.

**(A)**

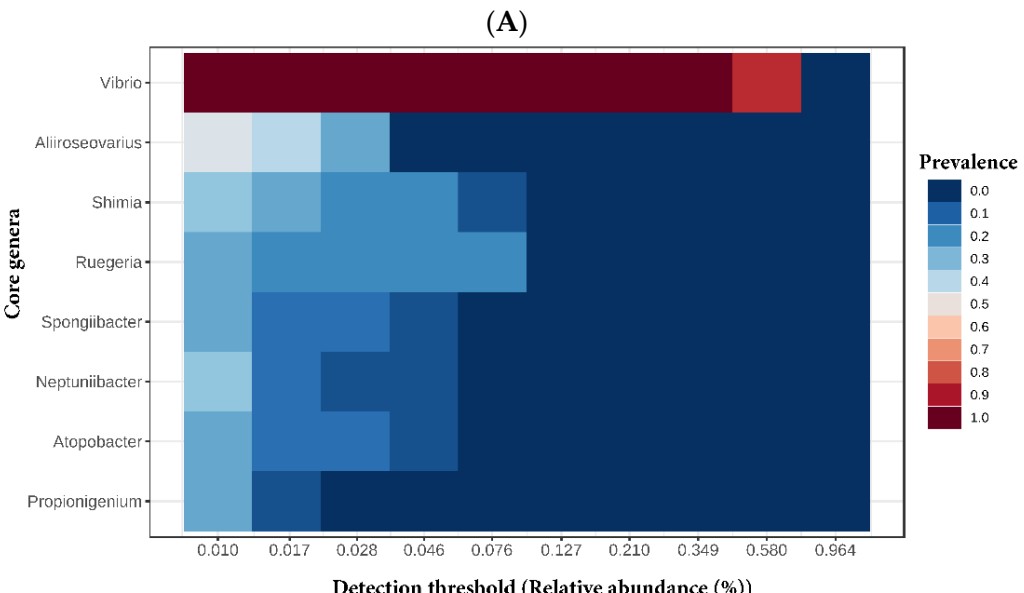

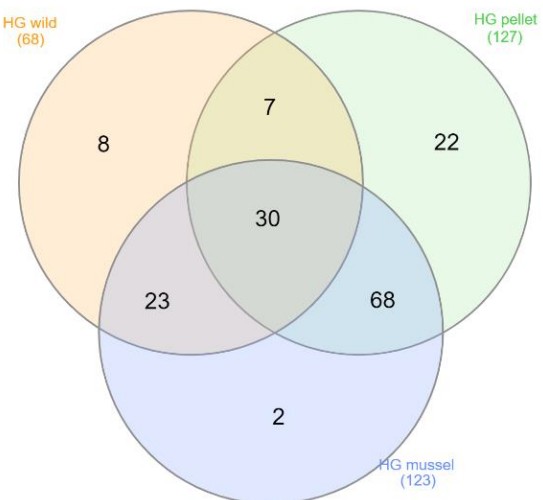

**Figure 4.** *Cont.*

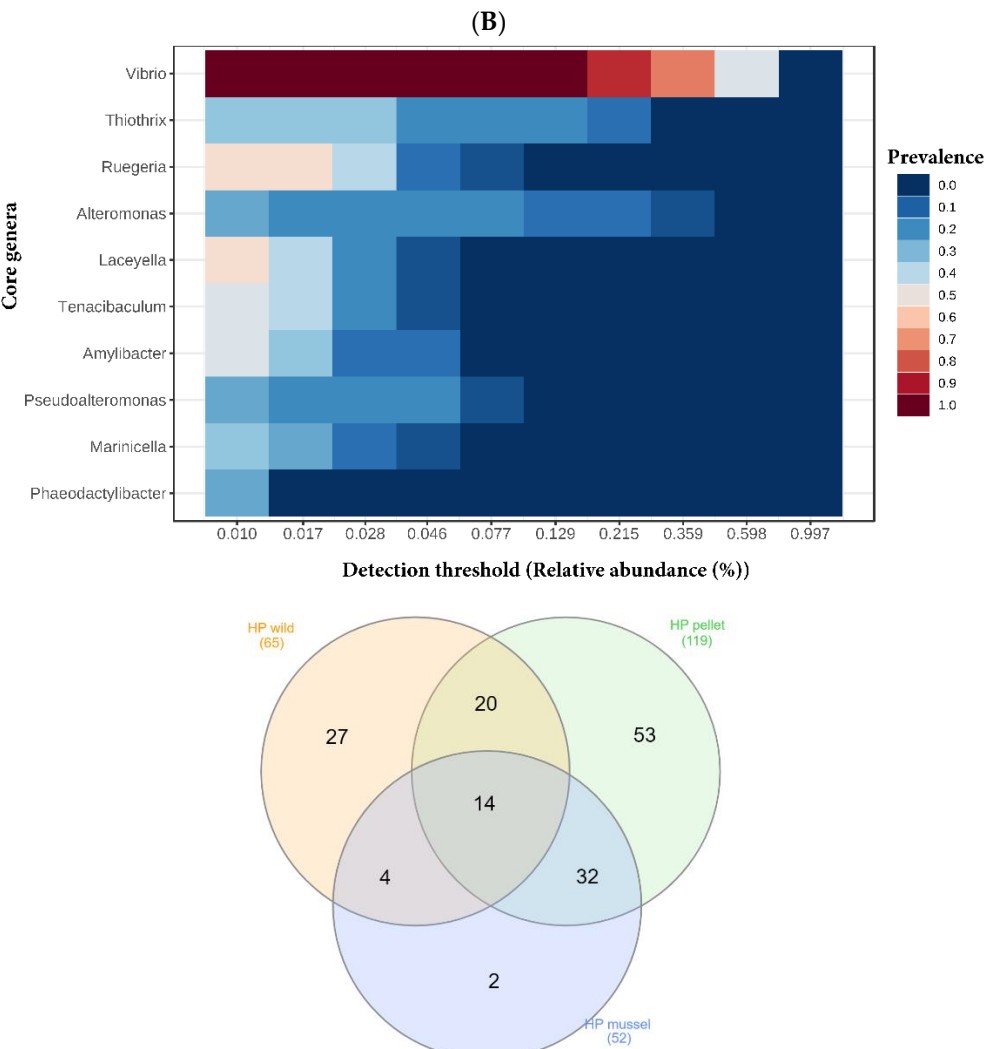

**Figure 4.** Core microbiota and Venn diagrams of bacterial genera in the (**A**) hindgut (*n* = 15) and (**B**) hepatopancreas (*n* = 15) of *P. ornatus* samples. Core microbiota heatmaps are colour coded, indicating the fraction of samples (prevalence) where each core genera (y-axis) is found at the relative abundance threshold shown on the x-axis.

## 4. Discussion

The hindgut and hepatopancreas microbiota of wild lobsters were significantly different to the cultured animals, as evidenced by PERMANOVA results and principal coordinate analysis plots. The tight clusters of wild *P. ornatus* mean that there was high microbial similarity among the individuals. However; the core microbiota in the hindgut and hepatopancreas of *P. ornatus* were the same regardless of the diet, and this is consistent with previous findings in *L. vannamei* [3]. The major phyla Proteobacteria and Bacteroidetes found in the hindgut in this study are consistent with results of the hindgut microbiota of younger *P. ornatus* juveniles previously identified in an earlier study from the same facility [9].

When the three treatment groups were compared, *Vibrio owensii* was highly abundant in both the hindgut and hepatopancreas of the mussel-fed lobsters. As a potential pathogen to crustaceans, *V. owensii* was initially isolated from the larvae of diseased cultured *P. ornatus* and black tiger shrimp *Penaeus monodon* [15]. This *Vibrio* species was associated with acute hepatopancreatic necrosis disease (AHPND) in the post-larvae of *L. vannamei* [16]. The findings suggest that fresh mussel may be a potential source of pathogens that can predispose lobsters to disease. Another aetiological agent of AHPND of *L. vannamei*, *Vibrio campbellii*, was found to be more common in the hepatopancreas of wild lobsters in this study. In

addition, *V. jasicida* was represented more consistently in the hindgut and hepatopancreas of pellet-fed animals. *Vibrio jasicida* (previously classified as *Vibrio harveyi*) is the infectious agent for luminous vibriosis in *Jasus verreauxi* lobster larvae [17]. Although the lobsters in this study were apparently healthy, the bacterial species mentioned above are opportunistic pathogens potentially causing disease when animals are immunocompromised. Vibrios such as *V. alginolyticus*, *V. fluvialis*, *V. furnissii*, *V. harveyi*, *V. parahaemolyticus* and *V. vulnificus* that are identified in this study have been known to cause zoonotic diseases in humans [18].

While some vibrios are pathogenic, there are other *Vibrio* species that form positive relationships with the host. Vibrios have been reported to have roles in inhibiting the growth of other bacteria, chitin and algal digestion in in vitro assays [19–21]. For example, *Vibrio hepatarius* detected in the hindgut and hepatopancreas of all treatment groups in this study may have probiotic properties [22,23]. Other instances of potential symbionts of lobsters are members of *Ruegeria* [24] and *Phaeobacter* [25] that can produce the antibacterial compound tropodithietic acid. Genus *Marinicella* present in the hepatopancreas of cultured lobsters in this study was also detected in the hindgut of *L. vannamei* [26], and may assist with denitrification by converting nitrate to nitrogen gas [27]. Similarly, the *Thiothrix* found in the cultured juveniles was also prevalent in *P. ornatus* phyllosomas [28]. Both *Marinicella* [27] and *Thiothrix* [28,29] are sulfur-oxidising bacteria, but their relationships with lobsters require further investigation. For instance, *Thiothrix* had been reported to cause filamentous bacterial disease in *P. ornatus* phyllosomas [30] and penaeid shrimp [31]. Genus *Thiothrix* was relatively more abundant in the hepatopancreas of pellet-fed lobsters compared to the mussel-fed animals in this study. The kuruma prawn pellet is made from meals with a higher protein composition (53%) than that of fresh mussel (41%). The higher relative abundance of *Thiothrix* in pellet-fed juveniles could be associated with its sulfur-oxidising role during protein digestion.

The microbiota of the cultured animals was more similar to each other compared to those of wild lobsters. This is shown in the principal coordinate analysis plot of the hepatopancreas and Venn diagrams of the hepatopancreas and hindgut. A possible explanation is that the culture environment has a strong influence as the cultured animals in this study were bred and grown in the same facility. The animals were fed with mussel, pellet and proprietary IMAS feed until weaned prior to treatment. [32] also reported no distinct difference in the midgut microbiota between reared Norway lobster *Nephrops norvegicus* fed with mussel or pellet. Nevertheless, there were several bacterial features that were significantly different in relative abundance between lobsters given pellet compared to those fed with mussel and the wild juveniles in the present study. These results are consistent with the findings of [4], which showed that diet places selective pressure on the gut bacteria of *L. vannamei*. In general, the diversity of gut microbiota can be affected by the trophic level of organisms that the animals feed on [33]. The wild lobsters are assumed to have fed on organisms belonging to several trophic levels available to them in the wild, while the mussel-fed lobsters were restricted to the blue mussel, a secondary consumer having consumed primary organisms such as phytoplankton and environmental bacteria. Furthermore, while the kuruma prawn pellet contains higher trophic organisms such as fish, squid, krill, shrimp and shellfish, these are unlikely to have transferred their associated microbiota to the lobsters due to the high temperature kuruma pellet manufacturing process. As no statistically significant differences were found among the alpha diversity indices between the three lobster feeds, it is unlikely that the trophic level of organisms within each feed greatly affected the microbial diversity of the lobsters.

As the hepatopancreas is a part of the midgut in which the digesta flows before reaching the hindgut, there were numerous microbial composition similarities between these two organs. These two organs were dominated by the phylum Proteobacteria, classes Gammaproteobacteria and Alphaproteobacteria, family *Vibrionaceae* and genus *Vibrio*. Moreover, similar trends were observed in the hepatopancreas and hindgut with significantly different taxonomic relative abundance among treatment groups in both organs such as the class Deltaproteobacteria, family *Carnobacteriaceae*, species *V. owensii*,

*V. jasicida* and *V. hyugaensis*. From the Venn diagrams of the hindgut and hepatopancreas libraries, the highest number of bacterial species belonged to the pellet-fed lobsters, while the mussel-fed animals had the lowest unique species number. On the other hand, older juveniles of cultured *P. ornatus* juveniles in an earlier study showed significant differences in bacterial composition between the midgut and hindgut [9,10] reported that the microbiota of the hindgut and hepatopancreas of *L. vannamei* were different regardless of the shrimp source. Microbiota in the midgut could have a more stable environment than the hindgut, which has sloughing of the innermost chitin layer during each moulting [8].

## 5. Conclusions

In conclusion, the hindgut and hepatopancreas microbiota of mussel-fed cultured lobsters was different from their wild counterparts. While the impact of the different gut microbiota between groups of lobsters was beyond the scope of this study, the differences observed suggests an opportunity to investigate the impact of different lobster gut microbiota on lobster performance and health. Furthermore, the high relative abundance of *V. owensii*, a known pathogen of the spiny lobster, in both the hindgut and hepatopancreas of the mussel-fed lobsters validates biosecurity concerns regarding feeding fresh mussel to the juvenile spiny lobster, and the importance of the development of manufactured feeds.

**Author Contributions:** A.R.B., A.J.T. and G.G.S. conceived and designed the experiments. M.C.O., A.J.T., A.R.B. and G.G.S. performed the experiments. M.C.O., A.R.B. and A.J.T. analysed the data. M.C.O., A.R.B., A.J.T. and G.G.S. wrote the paper. All authors have read and agreed to the published version of the manuscript.

**Funding:** This research was supported by the Australian Research Council Industrial Transformation Hub for Sustainable Onshore Lobster Aquaculture (project number IH90100014).

**Institutional Review Board Statement:** Not applicable.

**Informed Consent Statement:** Not applicable.

**Data Availability Statement:** The datasets generated and analysed during the current study are available in the NCBI SRA under BioProject accession number PRJNA821626.

**Conflicts of Interest:** The authors declare no conflict of interest.

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
