# Peer review of "Characterisation of the Gut Bacteria of Cultured and Wild Spiny Lobster Panulirus ornatus"

_2673-8007, doi:10.3390/applmicrobiol3010016_

Round 1
Reviewer 1 Report
Ooi et al. studied the hindgut and hepatopancreas bacterial diversity of wild spiny lobster.
Overall the study was well designed and interesting.
However I have some hesitations regarding the presentation and the final discussion.
Although results are clear the way they are being presented is quite tiring. I believe they should not be so explicitly written in text. Better figures could assist on that and just focus in striking differences.
Also Figure 4 is not clear to me (what do you mean with detection threshold?) the legends should be more clear.
Finally in the discussion apart from the potential pathogens detected I believe that the authors should focus more on trophic relationships and changes between different treatments.
Author Response
Point 1: Although results are clear the way they are being presented is quite tiring. I believe they should not be so explicitly written in text. Better figures could assist on that and just focus in striking differences.
Response 1: Figures 2 and 3 were focussed at family and genus levels. Hence the relative abundance at different taxonomic levels was described in the text.
Point 2: Also Figure 4 is not clear to me (what do you mean with detection threshold?) the legends should be more clear.
Response 2: The phrase ‘Detection Threshold’ was removed from Figure 4, leaving the x-axis as ‘Relative Abundance (%)’.
Point 3: Finally in the discussion apart from the potential pathogens detected I believe that the authors should focus more on trophic relationships and changes between different treatments.
Response 3: In the Discussion, Paragraph 3 described other possible relationships apart from potential pathogens:
While some vibrios are pathogenic, there are other Vibrio species that form positive relationships with the host. Vibrios have been reported to have roles in inhibiting growth of other bacteria, chitin and algal digestion in in vitro assays (Suginta et al., 2000, Wietz et al., 2010, Martin et al., 2014). For example, Vibrio hepatarius detected in the hindgut and hepatopancreas of all treatment groups in this study may have probiotic properties (Gullian et al., 2004, Ramirez et al., 2022). Other instances of potential symbionts of lobsters are members of Ruegeria (Brinkhoff et al., 2004) and Phaeobacter (Bruhn et al., 2007) that can produce the antibacterial compound tropodithietic acid. Genus Marinicella present in the hepatopancreas of cultured lobsters in this study was also detected in the hindgut of L. vannamei (Zheng et al., 2020), and may assist with denitrification by converting nitrate to nitrogen gas (Van Duc et al., 2018). Similarly, Thiothrix found in the cultured juveniles was also prevalent in P. ornatus phyllosomas (Payne et al., 2007). Both Marinicella (Van Duc et al., 2018) and Thiothrix (Brigmon & De Ridder, 1998, Payne et al., 2007) are sulfur-oxidising bacteria, but their relationships with lobsters require further investigation.
In the Discussion, changes between different treatments were discussed in Paragraph 4:
The microbiota of the cultured animals was more similar to each other compared to those of wild lobsters. This is shown in the principal coordinate analysis plot of the hepatopancreas and Venn diagrams of the hepatopancreas and hindgut. A possible explanation is that the culture environment has a strong influence as the cultured animals in this study were bred and grown in the same facility. The animals were fed with mussel, pellet and proprietary IMAS feed until weaned prior to treatment. Meziti et al. (2012) also reported no distinct difference in the midgut microbiota between reared Norway lobster Nephrops norvegicus fed with mussel or pellet. Nevertheless, there were several bacterial features that were significantly different in relative abundance between lobsters given pellet compared to those fed with mussel and the wild juveniles in the present study. These results are consistent with the findings of Qiao et al. (2016) which showed that diet places selective pressure on gut bacteria of L. vannamei.
Reviewer 2 Report
This paper is very important and have great significance. Traits of the gut bacteria of cultured and wild spiny lobster is different, which have a significant difference. In conclusion, the hindgut and hepatopancreas microbiota of mussel-fed cultured lobsters was different to their wild counterparts. While the impact of the different gut
microbiota between groups of lobsters was beyond the scope of this project the differences observed in this study provides the opportunity for further research to investigate the potential impacts of differences in lobster gut microbiota on lobster performance and health. Furthermore, the high relative abundance of V. owensii, a known pathogen of the spiny lobster, in both the hindgut and hepatopancreas of the mussel-fed lobsters validates biosecurity concerns regarding feeding fresh mussel to the juvenile spiny lobster, and the importance of manufactured feeds. That is, this paper provided a basis for bait research and may be supplied a way for lobster breeding. This paper can be accepted after a minor revision.
All figures need increased clarity.
Author Response
Point 1: All figures need increased clarity.
Response 1: All the figures have been replaced as pictures in TIFF and SVG formats to provide better clarity/resolution.
Reviewer 3 Report
First, I think several individuals' alpha diversities are not suitable for analysis. For example,the Simpson of W3 (0.42) which might be wrong with this sample, and filtered sequences of K2A2, et. Please check them.
Second, the results and discussions in this study are not adequate. In this study, the different family and genus of bacteria were displayed, but the reasons contributing to the difference were not fully discussed.
Last. Vibrio is the key and prodomint genus in all the sample. As we all known, Vibiro is a zoonotic bacteria, thus it is important to elucidate the species of Vibrio, and discuss the potential source.
Author Response
Point 1: First, I think several individuals' alpha diversities are not suitable for analysis. For example, the Simpson of W3 (0.42) which might be wrong with this sample, and filtered sequences of K2A2, et. Please check them.
Response 1:
In Table 1, the phrase ‘filtered sequences’ was replaced with ‘original read counts’ to avoid confusion regarding the reads used to calculate the alpha diversity indices. Alpha diversity indices were calculated after the original read counts were filtered to a minimum count of 2 with 10% prevalence in the sample, and normalised by rarefying to the minimum library size and total sum scaling as described in the Materials and Methods. A note was also included in the table caption indicating that normalised counts and not the original counts were used to calculate the observed features, Shannon and Simpson alpha diversity indices.
Section 2.5 of Materials and Methods:
Data were filtered to a minimum count of 2 with 10% prevalence in the sample and nor-malised by rarefaction to the minimum library size (i.e. 14485 for hindgut and 8182 for hepatopancreas) and total sum scaling.
Table 1. Alpha diversity indices for normalised gut libraries of juvenile P. ornatus.
Point 2: Second, the results and discussions in this study are not adequate. In this study, the different family and genus of bacteria were displayed, but the reasons contributing to the difference were not fully discussed.
Response 2:
The reasons contributing to the differences between treatments were discussed in Paragraphs 4 and 5 in the Discussion section:
The microbiota of the cultured animals was more similar to each other compared to those of wild lobsters. This is shown in the principal coordinate analysis plot of the hepatopancreas and Venn diagrams of the hepatopancreas and hindgut. A possible explanation is that the culture environment has a strong influence as the cultured animals in this study were bred and grown in the same facility. The animals were fed with mussel, pellet and proprietary IMAS feed until weaned prior to treatment. Meziti et al. (2012) also reported no distinct difference in the midgut microbiota between reared Norway lobster Nephrops norvegicus fed with mussel or pellet. Nevertheless, there were several bacterial features that were significantly different in relative abundance between lobsters given pellet compared to those fed with mussel and the wild juveniles in the present study. These results are consistent with the findings of Qiao et al. (2016) which showed that diet places selective pressure on gut bacteria of L. vannamei.
As the hepatopancreas is a part of the midgut in which the digesta flows before reaching the hindgut, there were numerous microbial composition similarities between these two organs. These two organs were dominated by phylum Proteobacteria, classes Gammaproteobacteria and Alphaproteobacteria, family Vibrionaceae and genus Vibrio. Moreover, similar trends were observed in the hepatopancreas and hindgut with significantly different taxonomic relative abundance among treatment groups in both organs such as class Deltaproteobacteria, family Carnobacteriaceae, species V. owensii, V. jasicida and V. hyugaensis. From the Venn diagrams of the hindgut and hepatopancreas libraries, the highest number of bacterial species belonged to the pellet-fed lobsters while the mussel-fed animals had the lowest unique species number. On the other hand, older juveniles of cultured P. ornatus juveniles in an earlier study showed significant differences in bacterial composition between the midgut and hindgut (Ooi et al., 2017). Cornejo-Granados et al. (2017) reported microbiota of the hindgut and hepatopancreas of L. vannamei were different regardless of the shrimp source. Microbiota in the midgut could have a more stable environment than the hindgut which has sloughing of the innermost chitin layer during each moulting (Ceccaldi, 1989).
Point 3: Last. Vibrio is the key and predominant genus in all the sample. As we all known, Vibiro is a zoonotic bacteria, thus it is important to elucidate the species of Vibrio, and discuss the potential source.
Response 3:
Potential zoonotic vibrios were added to the Discussion:
Vibrios such as V. alginolyticus, V. fluvialis, V. furnissii, V. harveyi, V. parahaemolyticus and V. vulnificus identified in this study, have been known to cause zoonotic diseases in humans (Austin, 2010).
Point 4: Areas that can be improved - Introduction, Research design, Results and Conclusions.
Response 4: Some changes to the language were made in the Introduction, Materials and Methods, Discussion and Conclusions sections to improve the manuscript. Please refer to the track changes in the manuscript.
Round 2
Reviewer 1 Report
The authors tried to reply to most of the comments.
I still do not understand figure 4 and I find that the legend does not include the necessary information.
a) what is prevalence?
b) what is axis x showing? relative abundance of the species? where? in which sample?
c) why do you have the same figure 2 times? I think they are exactly the same.
I still believe that the authors should try toe explain these significant differences they found between pellet and mussel fed lobsters microbiota in terms of trophic relationships (i.e. more protein in mussels? Can we see something that explains that?)
Author Response
I still do not understand figure 4 and I find that the legend does not include the necessary information.
Point 1: a) what is prevalence?
Figure 4 includes heatmaps representing the core microbiota of the two organs (15 samples/libraries of each organ). This form of graphical visualisation is standard for many microbiome studies and is a heatmap where the prevalence (standard meaning-fraction of total samples) at several relative abundance thresholds of each genus representing the core microbiota are colour coded from low (shades of blue) to high (dark red).
Point 2: b) what is axis x showing? relative abundance of the species? where? in which sample?
Response 2: The axis x shows the relative abundance of the bacterial genus in each organ. For example, the first row in Figure 4A shows that 100% (1.0 prevalence) of all the hindgut samples had 45% relative abundance (0.45) of Vibrio. The heatmap also shows that when Vibrio sp. occur at greater than 45% relative abundance this is found in fewer samples (reduced prevalence) hence the lighter shade of red.
The caption for Figure 4 has been edited:
Figure 4. Core microbiota and Venn diagrams of bacterial genera in the (A) hindgut (n=15) and (B) hepatopancreas (n=15) P. ornatus libraries. Core microbiota heatmaps are colour coded indicating the fraction of samples (prevalence) where each core genus (y-axis) is found at the relative abundance shown on the x-axis.
Point 3: c) why do you have the same figure 2 times? I think they are exactly the same.
Response 3: The previous core microbiota figures were deleted (as x-axes were changed), but still visible as track changes were used. The current version only has a core microbiota figure for each sample type.
Point 4: I still believe that the authors should try to explain these significant differences they found between pellet and mussel fed lobsters microbiota in terms of trophic relationships (i.e. more protein in mussels? Can we see something that explains that?)
Response 4:
The kuruma prawn pellet has a higher protein composition than fresh mussel. It is mentioned in the Discussion section (Paragraph 3):
While some vibrios are pathogenic, there are other Vibrio species that form positive relationships with the host. Vibrios have been reported to have roles in inhibiting growth of other bacteria, chitin and algal digestion in in vitro assays (Suginta et al., 2000, Wietz et al., 2010, Martin et al., 2014). For example, Vibrio hepatarius detected in the hindgut and hepatopancreas of all treatment groups in this study may have probiotic properties (Gullian et al., 2004, Ramirez et al., 2022). Other instances of potential symbionts of lobsters are members of Ruegeria (Brinkhoff et al., 2004) and Phaeobacter (Bruhn et al., 2007) that can produce the antibacterial compound tropodithietic acid. Genus Marinicella present in the hepatopancreas of cultured lobsters in this study was also detected in the hindgut of L. vannamei (Zheng et al., 2020), and may assist with denitrification by converting nitrate to nitrogen gas (Van Duc et al., 2018). Similarly, Thiothrix found in the cultured juveniles was also prevalent in P. ornatus phyllosomas (Payne et al., 2007). Both Marinicella (Van Duc et al., 2018) and Thiothrix (Brigmon & De Ridder, 1998, Payne et al., 2007) are sulfur-oxidising bacteria, but their relationships with lobsters require further investigation. For instance, Thiothrix had been reported to cause filamentous bacterial disease in P. ornatus phyllosomas (Bourne et al., 2007) and penaeid shrimp (Bower et al., 1994). Genus Thiothrix was relatively more abundant in the hepatopancreas of pellet-fed lobsters compared to the mussel-fed animals in this study. The kuruma prawn pellet is made from meals with a higher protein composition (53%) than that of fresh mussel (41%). The higher relative abundance of Thiothrix in pellet-fed juveniles could be associated with its sulfur-oxidising role during protein digestion.
Trophic relationships are added to the Discussion section (Paragraph 4):
The microbiota of the cultured animals was more similar to each other compared to those of wild lobsters. This is shown in the principal coordinate analysis plot of the hepatopancreas and Venn diagrams of the hepatopancreas and hindgut. A possible explanation is that the culture environment has a strong influence as the cultured animals in this study were bred and grown in the same facility. The animals were fed with mussel, pellet and proprietary IMAS feed until weaned prior to treatment. Meziti et al. (2012) also reported no distinct difference in the midgut microbiota between reared Norway lobster Nephrops norvegicus fed with mussel or pellet. Nevertheless, there were several bacterial features that were significantly different in relative abundance between lobsters given pellet compared to those fed with mussel and the wild juveniles in the present study. These results are consistent with the findings of Qiao et al. (2016) which showed that diet places selective pressure on gut bacteria of L. vannamei. In general, the diversity of gut microbiota can be affected by the trophic level of organisms that the animals feed on (Liu et al., 2016). The wild lobsters are assumed to have fed on organisms belonging to several trophic levels available to them in the wild while the mussel-fed lobsters were restricted to the blue mussel, a secondary consumer having consumed primary organisms such as phytoplankton and environmental bacteria. Furthermore, while the kuruma prawn pellet contains higher trophic organisms such as fish, squid, krill, shrimp and shellfish, these are unlikely to have transferred their associated microbiota to the lobsters due to the high temperature kuruma pellet manufacturing process. As no statistically significant differences were found among the alpha diversity indices between the three lobster feeds it is unlikely that the trophic level of organisms within each feed greatly affected the microbial diversity of the lobsters.